# Epidemiological and health economic implications of symptom propagation in respiratory pathogens: A mathematical modelling investigation

**Phoebe Asplin** [1,2,3]*, **Matt J. Keeling** [2,3,4], **Rebecca Mancy** [5,6], **Edward M. Hill** [2,3]*

**1** EPSRC & MRC Centre for Doctoral Training in Mathematics for Real-World Systems, University of Warwick, Coventry, United Kingdom, **2** Mathematics Institute, University of Warwick, Coventry, United Kingdom, **3** The Zeeman Institute for Systems Biology & Infectious Disease Epidemiology Research, University of Warwick, Coventry, United Kingdom, **4** School of Life Sciences, University of Warwick, Coventry, United Kingdom, **5** School of Biodiversity, One Health and Veterinary Medicine, University of Glasgow, Glasgow, United Kingdom, **6** MRC/CSO Social and Public Health Sciences Unit, University of Glasgow, Glasgow, United Kingdom

\* P.Asplin@warwick.ac.uk (PA); Edward.Hill@warwick.ac.uk (EMH)

**Data Availability Statement:** The model is parameterised using estimates from the literature, as described in the main text, with relevant references provided. The code repository for the

## Abstract

### Background

Respiratory pathogens inflict a substantial burden on public health and the economy. Although the severity of symptoms caused by these pathogens can vary from asymptomatic to fatal, the factors that determine symptom severity are not fully understood. Correlations in symptoms between infector-infectee pairs, for which evidence is accumulating, can generate large-scale clusters of severe infections that could be devastating to those most at risk, whilst also conceivably leading to chains of mild or asymptomatic infections that generate widespread immunity with minimal cost to public health. Although this effect could be harnessed to amplify the impact of interventions that reduce symptom severity, the mechanistic representation of symptom propagation within mathematical and health economic modelling of respiratory diseases is understudied.

### Methods and findings

We propose a novel framework for incorporating different levels of symptom propagation into models of infectious disease transmission via a single parameter, $\alpha$. Varying $\alpha$ tunes the model from having no symptom propagation ($\alpha = 0$, as typically assumed) to one where symptoms always propagate ($\alpha = 1$). For parameters corresponding to three respiratory pathogens—seasonal influenza, pandemic influenza and SARS-CoV-2—we explored how symptom propagation impacted the relative epidemiological and health-economic performance of three interventions, conceptualised as vaccines with different actions: symptom-attenuating (labelled SA), infection-blocking (IB) and infection-blocking admitting only mild breakthrough infections (IB_MB).

study is available at: https://github.com/pasplin/symptom-propagation-mathematical-modelling. Archived code: https://doi.org/10.5281/zenodo.10560986.

**Funding:** PA and MJK were supported by the Engineering and Physical Sciences Research Council through the MathSys CDT [grant number EP/S022244/1]. MJK was also supported by the Medical Research Council through the JUNIPER partnership award [grant number MR/X018598/1]. MJK and EMH are linked with the JUNIPER partnership [MRC grant number MR/X018598/1] and would like to acknowledge their help and support. RM was supported by The Leckie Fellowship, the Medical Research Council [grant number MC_UU_00022/4] and the Chief Scientist Office [grant number SPHSU19]. The funders had no role in study design, data collection and analysis, decision to publish, or preparation of the manuscript.

**Competing interests:** The authors have declared that no competing interests exist.

In the absence of interventions, with fixed underlying epidemiological parameters, stronger symptom propagation increased the proportion of cases that were severe. For SA and IB_MB, interventions were more effective at reducing prevalence (all infections and severe cases) for higher strengths of symptom propagation. For IB, symptom propagation had no impact on effectiveness, and for seasonal influenza this intervention type was more effective than SA at reducing severe infections for all strengths of symptom propagation. For pandemic influenza and SARS-CoV-2, at low intervention uptake, SA was more effective than IB for all levels of symptom propagation; for high uptake, SA only became more effective under strong symptom propagation. Health economic assessments found that, for SA-type interventions, the amount one could spend on control whilst maintaining a cost-effective intervention (termed threshold unit intervention cost) was very sensitive to the strength of symptom propagation.

## Conclusions

Overall, the preferred intervention type depended on the combination of the strength of symptom propagation and uptake. Given the importance of determining robust public health responses, we highlight the need to gather further data on symptom propagation, with our modelling framework acting as a template for future analysis.

### Author summary

Symptom propagation occurs when the symptoms of an infected individual depend, at least partially, on the symptoms of the person who infected them. An example is when an individual is more likely to develop severe symptoms if infected by someone with severe symptoms themselves. Symptom propagation has important implications for infection control strategies and could be harnessed to amplify the impact of vaccines that reduce the probability of severe disease.

Evidence for symptom propagation is growing, yet it is rarely included in models of infectious disease transmission. Here, we provide a new infectious disease transmission model with a single parameter representing the strength of symptom propagation and study the consequences for vaccination control strategies. We show that the strength of symptom propagation has profound effects on infectious disease outbreaks, including notably on the proportion of cases that are severe.

We demonstrate that vaccines that reduce symptom severity are more effective in reducing severe and overall cases when symptom propagation is stronger. Knowing the strength of symptom propagation can help understand the effectiveness of vaccines that reduce the risk of infection relative to those that reduce symptoms, helping to shape public health strategy.

## Introduction

Respiratory pathogens, of which influenza and SARS-CoV-2 are prominent examples, are those that cause infection in the respiratory tract, and are a major cause of mortality worldwide in high, medium and low income countries [1]. Many respiratory pathogens have demonstrated their capability to cause large-scale epidemics and/or pandemics. For example, seasonal

influenza causes annual epidemics which, prior to the COVID-19 pandemic beginning in 2020, were estimated to result in symptomatic infection of 8% of the US population each year on average [2] and around 290,000 to 650,000 deaths globally [3]. Pandemic influenza has also inflicted devastating consequences on global public health; the 1918/19 Spanish flu pandemic is thought to have resulted in 50 million deaths worldwide [4], while the 2009 H1N1 pandemic caused 200,000 deaths in its first year of circulation [5]. Since its emergence in humans in 2019, SARS-CoV-2, the causative agent of COVID-19 disease, has resulted in an estimated number of global deaths exceeding 6.5 million by the end of 2022 [6].

Whilst the serious public health risks posed by respiratory diseases are evident, the resulting outbreaks also come with a considerable economic cost. COVID-19 has had a massive impact on the global economy, with the global cost in 2020 and 2021 estimated to be 14% of 2019 GDP [7]. These alarming valuations were partially due to the high cost of interventions. For example, by September 2021, the UK had spent £17.9bn on the test and trace programme, £13.8bn on the procurement of personal protective equipment and £1.8bn on vaccine and antibody supply [8].

Although the headline statistics on the health burden of these pathogens are somewhat bleak, many respiratory pathogens are capable of causing of range of symptomatic outcomes. Often, infected individuals experience, at worst, only mild symptoms, such as a runny nose, focused in the upper respiratory tract. On the other hand, such pathogens also have the potential to cause severe symptoms, primarily by infecting lower parts of the respiratory tract [9, 10]. Indeed, lower respiratory tract infections account for more than 2.4 million deaths worldwide each year [1] and are a leading cause of global mortality, especially amongst children and elderly people [11].

The COVID-19 pandemic has heralded a paradigm shift in modelling the actions of interventions when assessing public health control strategies, highlighting the importance of symptom severity and leading to an increased understanding of the action of interventions beyond being purely infection-blocking [12, 13]. Vaccines, whilst previously viewed in modelling terms as solely a way to prevent infection, are now being considered to have a dual action of reducing symptom severity [14–16]. Similarly, non-pharmaceutical interventions such as mask-wearing, social distancing and hand washing were previously only thought of as ways to reduce transmission but are now thought to additionally reduce the likelihood of symptomatic infection [17–21].

Preparedness efforts against respiratory disease outbreaks and the contemporary evaluations of intervention effectiveness motivate research into the relationship between the severity of illness, viral load and the transmission routes through which they spread. This research has revealed good biological grounds for investigating symptom propagation for respiratory pathogens, where we define symptom propagation as occurring when the symptoms of an infected individual depend, at least partially, on the symptoms of their infector. Symptom propagation has been documented for *Yersinia pestis*, the causative agent of plague. Those who develop the more severe form, pneumonic plague, are then able to infect via the aerosolised route, which results in pneumonic plague in those infected [22].

There is growing evidence that symptom propagation occurs for other respiratory pathogens [23]. Prior studies of influenza and SARS-CoV-2, have suggested two pathways through which symptoms may propagate along chains of infection [24, 25]. The first pathway is through a dose-response relationship. Individuals presenting with severe disease tend to shed more viral particles [26–29], meaning that those they infect receive a larger infectious dose, in turn increasing the probability of more severe disease outcomes [25, 30–32]. The second pathway is through differential transmission routes: it is thought that severe disease arises more frequently for aerosol transmission (transmission involving particles smaller than $5\mu m$, which are

sufficiently small and light to travel on air flows and to enter the lower respiratory tract) than for close contact transmission (transmission involving direct or indirect contact with an infected individual or transmission via large droplets, which are more likely to lodge in the upper respiratory tract) [33]. The association with severity arises because aerosol transmission is more likely to cause infection in the lower respiratory tract, resulting in a higher probability of more severe disease [24, 31, 34]. These studies give evidence for symptom propagation between infector-infectee pairs, but its incorporation into epidemiological models is required to fully appreciate its importance at a larger scale.

Symptom propagation has the potential to create chains of severe infections, resulting in large, intensive outbreaks that could have devastating consequences for groups most at risk; on the other hand, it could result in mild or asymptomatic infection spreading through a population, creating widespread immunity whilst incurring a minimal cost to public health. In light of the observed differential symptom severity that may be experienced for many pathogens, we are interested in exploring the public health ramifications of a relationship between symptom severity of an infected individual and the symptom severity of any subsequent cases caused by onward transmission.

We view the absence of a mechanistic representation of the propagation of symptom sets as a modelling 'blind spot', in light of the earlier described biological evidence already giving strong support of it being a notable process for some pathogens [22]. Symptom severity has typically been modelled post-hoc or separately from epidemiological dynamics. For example, it has become commonplace for models to distinguish between asymptomatic and symptomatic infection, but asymptomatic infections are generally assumed to occur with a fixed probability, independent of other infected individuals [35]. An extension to this model has been explored in Paulo et al. [36], where the probability of severe disease depended on the proportion of the population infected at the time, although not on their severity. Other models in the literature capture multi-route transmission but do not invoke a relationship between the route of transmission and symptom severity [24, 37–39]. Initial attempts to incorporate symptom propagation into an epidemiological model of a respiratory tract infection can be seen in Earnest [40] and Harris et al. [41]; however, work in this area remains rudimentary.

Another aspect meriting greater attention is the impact of symptom propagation on health economic outcomes used to assess cost-effectiveness and help optimise public health strategies. For seasonal influenza, there has been a focus in previous health economic studies on the cost-effectiveness of vaccination scenarios [42, 43]. On the other hand, in the context of COVID-19, although health economic modelling studies have predominantly focused on vaccine roll-out [44], there have been evaluations of symptom-dependent interventions, such as comparing the effectiveness of symptomatic versus asymptomatic testing [45] and considering quarantining measures that predominantly target symptomatic individuals [46]. At the time of writing, no work has been done to explore the effect of symptom propagation on health economic outcomes.

In this paper, we develop a mathematical modelling framework that incorporates symptom propagation and apply it to a range of pathogens to investigate the epidemiological and health-economic implications of symptom propagation. First, we develop a generalisable, mechanistic infectious disease transmission model that incorporates different strengths of symptom propagation via a single parameter, which we call $\alpha$. Parameterising this model to capture three representative respiratory tract pathogens of public health concern (seasonal influenza, pandemic influenza and SARS-CoV-2), we conduct numerical experiments to explore the impacts of symptom propagation of different strengths on epidemiological and health economic outcomes, both with no intervention and under three interventions that we conceptualise as vaccines with different modes of action. As well as symptom severity propagation having

important impacts on natural epidemiological dynamics (in the absence of intervention), we found that when interventions were applied, symptom propagation acted to amplify the beneficial effects of symptom-attenuating interventions on community-level epidemiological outcomes. These effects became even more stark for cost-effectiveness based assessments. For pathogens where the propagation of symptom severity is an important attribute, our findings motivate the development and use of a new class of models to help identify the most appropriate type of intervention to harness the beneficial attributes of symptom propagation, delivering a reduced burden on public health and more cost-effective control policy.

## Methods

The propagation of symptom severity has been largely neglected in epidemiological modelling, with symptom severity having typically been modelled post-hoc or separately from epidemiological dynamics. With an application to outbreaks of respiratory pathogens of public health concern (namely seasonal influenza, pandemic influenza and SARS-CoV-2), we investigated the impact of the strength of symptom propagation on epidemiological and health economic measures. Our methodology comprised multiple aspects that we detail in turn: (i) a mathematical model of infectious disease transmission that included two symptom severity classes (mild and severe) and a mechanism for symptom propagation; (ii) incorporation of interventions into the model, with our implementation roughly corresponding to three plausible modes of action of a vaccine; (iii) approaches to assess the health economic implications of proposed strategies.

### Infectious disease model including symptom severity and propagation

The basis of our infectious disease model is a standard deterministic, compartmental susceptible-exposed-infectious-recovered (SEIR) model, described by a system of ordinary differential equations (ODEs) [47, 48]. We extended the framework by stratifying infections according to two levels of symptom severity, mild and severe, and mechanistically incorporating (the potential for) symptom propagation (Fig 1).

We expand below on: (a) the reasoning for the selection of two symptom severity classes; (b) a description of how symptom severity was augmented into the compartmental framework; (c) the transmission dynamics; (d) the mathematical formulation of the symptom propagation mechanism; (e) the collection of ODEs describing the system dynamics (in the absence of interventions); (f) the computational simulations performed to consider the implications of symptom propagation on our three exemplar pathogens (seasonal influenza, pandemic influenza, SARS-CoV-2) in the absence of interventions.

**(a) Two symptom severity classes.** No formal scheme exists for classifying disease severity, with differing interpretations possible across pathogens. For some pathogens, cases are typically stratified into 'mild' or 'severe'; in the context of influenza, severe disease is generally associated with the development of a cough or fever [35, 49, 50]. For other pathogens, severity is commonly stratified according to 'asymptomatic' or 'symptomatic' infection, with SARS-CoV-2 being a notable example [41, 51]. Due to the ubiquity of 'mild' and 'severe' within the literature for respiratory pathogens, we have decided to use this terminology throughout. It should be noted, however, that the parameters chosen for mild COVID-19 disease, such as the infectious period, are taken from estimates for the asymptomatic parameters. We model the propagation of mild and severe symptoms because of the clinical importance of symptom severity for respiratory tract infections but note that the framework can be applied to the propagation of symptom sets in general.

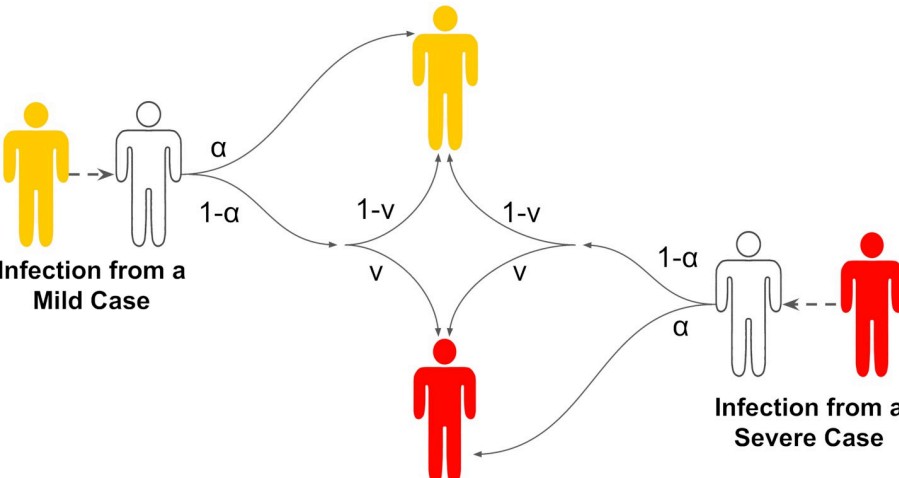

**Fig 1. Schematic showing how symptom severity was determined according to the two symptom severity associated parameters, *α* and *ν*.** White shaded individuals correspond to those susceptible to infection, yellow shaded individuals correspond to infectious cases with mild severity and red shaded individuals correspond to infectious cases with severe symptoms. The values on the arrows show the corresponding probability. In brief, an infected individual has probability $\alpha$ of copying the symptom severity of their infector and a probability $1 - \alpha$ of reverting to the baseline probability of having severe disease, i.e. they developed severe disease with probability $\nu$.

**(b) Infectious disease model compartments.** Under our model structure of compartmentalising the population into susceptible, exposed (infected but not yet infectious), infectious and recovered states, we further separated each of the exposed, infectious and recovered states into two classes representing the two symptom severity levels ('mild' and 'severe'). In detail, $E_M$ ($E_S$) contains individuals exposed to the disease who would go on to develop mild (severe) disease. $I_M$ ($I_S$) contains individuals who had become infectious and exhibited mild (severe) symptoms. Note that we assumed there was no movement between the severity classes, meaning an individual's severity would be constant across their exposed and infectious periods. $R_M$ ($R_S$) contains individuals who had experienced mild (severe) disease and since recovered.

**(c) Transmission dynamics.** We assumed a dependence between disease severity and both the rate of transmission of infection, $\beta_M$, $\beta_S$ (for 'mild' and 'severe' cases, respectively), and the recovery rate from infection, $\gamma_M$, $\gamma_S$. On the other hand, the incubation period, and thus the rate of becoming infectious, $\epsilon$, took the same fixed value for both severity classes due to data suggesting there is limited variation between mild and severe cases [50, 51].

Additionally, we assumed there was no waning immunity after recovery and we ignored demographic processes (natural births and deaths)—as such we are modelling a single epidemic outbreak. In the majority of scenarios, the simulated outbreaks occurred over a short time frame where the impacts of these waning and demographic processes would be negligible. Equally, in the unusual case that outbreaks persisted over a sustained time period (a decade and above), for the purposes of simplicity, we wanted to maintain the focus on the impact of symptom propagation-associated factors.

We explored three disease parameter sets (Table 1), chosen to reflect a range of disease scenarios: influenza-like parameters with $R_0 = 1.5$ (seasonal influenza), influenza-like parameters with $R_0 = 3.0$ (pandemic influenza) and early SARS-CoV-2-like parameters with $R_0 = 3.0$.

**Table 1. Epidemiological parameter values for the (seasonal and pandemic) influenza and SARS-CoV-2 scenarios.** In each parameterisation, we calibrated the $\beta$ value to acquire the stated value of $R_0$ in the respective scenario. All rates have a unit of 'per day' (day$^{-1}$). **(Top)** Parameters used for the two influenza scenarios: influenza-like parameters with $R_0 = 1.5$ (seasonal influenza parameterisation), influenza-like parameters with $R_0 = 3.0$ (pandemic influenza parameterisation). **(Bottom)** SARS-CoV-2-like parameters with $R_0 = 3.0$.

| Parameter | Description | Influenza value (day$^{-1}$) | Source |
|---|---|---|---|
| $\beta_M$ | Mild transmission rate | $\beta$ | |
| $\beta_S$ | Severe transmission rate | $2\beta$ | Couch et al. [26] |
| $\epsilon$ | Rate of becoming infectious | 1/2 | Cowling et al. [50] |
| $\gamma_M$ | Mild recovery rate | 1/5 | Cao et al. [61] |
| $\gamma_S$ | Severe recovery rate | 1/7 | Cao et al. [61] |

| Parameter | Description | SARS-CoV-2 value (day$^{-1}$) | Source |
|---|---|---|---|
| $\beta_M$ | Mild transmission rate | $\beta$ | |
| $\beta_S$ | Severe transmission rate | $4\beta$ | Letizia et al. [62] |
| $\epsilon$ | Rate of becoming infectious | 1/5 | Byrne et al. [51] |
| $\gamma_M$ | Mild recovery rate | 1/7 | Byrne et al. [51] |
| $\gamma_S$ | Severe recovery rate | 1/14 | Byrne et al. [51] |

Specifically, $R_0 = 1.5$ represented a pathogen that, on average, would spread through a population slowly and require minimal interventions to be suppressed. In contrast, $R_0 = 3.0$ represented a highly transmissible pathogen with the potential to infect the majority of the population in the absence of strong interventions. These values of $R_0$ were chosen to reflect estimates in the literature of 1–1.69 for seasonal influenza [52–54], 1.95–3.5 for pandemic influenza [55–57] and 2.43–3.60 for wild-type SARS-CoV-2 [58–60]. To obtain these fixed values of $R_0$, we computed the required value of the transmission rate $\beta$ for each value of $\alpha$ by deriving an equation for $R_0$ using the next-generation matrix approach (see Section 1 in S1 Text). To better align with approaches taken when analysing real-world infections, we chose to fix $R_0$ as opposed to fixing the value of $\beta$; $R_0$ is the parameter most likely to be known from available empirical measurements, with other model parameters (in this case $\beta$) inferred to generate the measured $R_0$ value. Our analysis with the value of $\beta$ fixed instead of $R_0$ can be found in Section 2 in S1 Text (Figs A and B).

Estimates for the other parameters were taken from studies of influenza A virus strains [26, 50, 61] and estimates for wild-type SARS-CoV-2 [51, 62]. Comparing between the pandemic influenza and SARS-CoV-2 parameter sets, the notable differences were SARS-CoV-2 having a higher ratio between mild and severe transmission rates (four for SARS-CoV-2, two for pandemic influenza), a longer incubation period (five days for SARS-CoV-2 versus two days for pandemic influenza) and a longer duration of infection for both mild and severe cases (full details in Table 1).

**(d) Incorporation of symptom propagation into the model framework.** We encapsulated symptom severity and symptom propagation into the model framework through two key parameters: $\alpha$—the dependence on the symptom severity of the infector, or equivalently, the strength of symptom propagation; $\nu$—the baseline probability of the pathogen causing severe disease in the absence of propagation effects. The parameter $\nu$ is aligned with the idea of 'virulence' in that it is a measure of the intrinsic severity of the pathogen. When $\alpha = 0$, the symptom severity of the infected individual has no dependence on the infector's symptom severity; instead, the symptom severity of the infected individual depends entirely on $\nu$—this corresponds to the typical assumption applied to compartmental infectious disease models. When $\alpha$

= 1, the symptom severity of an infected individual is wholly dependent on that of their infector, meaning that symptom severity is always passed on with infection; in the case of no interventions, this parameterisation is akin to a two-strain model, where one strain causes mild infections only and the other strain causes severe infections only. When $\alpha \in (0, 1)$, the symptom severity of the infectee has a partial dependence on the infector's symptom severity and a partial dependence on $v$.

Overall, our model of symptom propagation means that an infected individual, with probability $\alpha$, copies the symptom severity of their infector, whilst with probability $1 - \alpha$, their symptom severity is assigned randomly according to the underlying probability of having severe disease, $v$ (as depicted in Fig 1).

**(e) Baseline model equations (without interventions).** The rate of change for each disease state was governed by the system of differential equations shown in Eq 1, with parameters as described in Table 1 and Fig 1. This system of ODEs captures the different levels of disease severity, the dependence of the infectee's symptom severity on the infector's symptom severity (through the $\alpha$ parameter) and a baseline probability of an infected case having severe disease (through the $v$ parameter).

$$
\begin{aligned}
\frac{dS}{dt} &= -(\lambda_M + \lambda_S)S \\
\frac{dE_M}{dt} &= ((\alpha + (1-\alpha)(1-v))\lambda_M + (1-\alpha)(1-v)\lambda_S)S - \epsilon E_M \\
\frac{dE_S}{dt} &= ((1-\alpha)v\lambda_M + (\alpha + (1-\alpha)v)\lambda_S)S - \epsilon E_S \\
\frac{dI_M}{dt} &= \epsilon E_M - \gamma_M I_M \\
\frac{dI_S}{dt} &= \epsilon E_S - \gamma_S I_S \\
\frac{dR_M}{dt} &= \gamma_M I_M \\
\frac{dR_S}{dt} &= \gamma_S I_S,
\end{aligned}
\tag{1}
$$

where the force of infection from mild cases, $\lambda_M$, and severe cases, $\lambda_S$, respectively, were given by:

$$
\lambda_M = \frac{\beta_M I_M}{N}, \quad \lambda_S = \frac{\beta_S I_S}{N},
\tag{2}
$$

where N is the population size that is assumed to be constant.

**(f) Exploring the effect of symptom propagation on epidemiological dynamics in the absence of interventions: Simulation overview.** In all our model simulations described here and throughout the manuscript, we considered an outbreak arising within a population comparable in size to that of the UK ($N$ = 67 million). All the scenarios began with one infectious individual. We assumed that this individual would have severe symptoms with probability given by the baseline probability of severe disease ($v$). As the model used was fully deterministic, we chose to simulate this effect by splitting the single infectious individual between the two symptom severity classes so that $I_S$ was initialised to contain $v$ people and $I_M$ contained $1 - v$ people. The remainder of the population were initially susceptible (in the $S$ class). The

simulations were run until there was less than one individual combined across all the infected classes ($E_M$, $E_S$, $I_M$ and $I_S$).

All code was produced using Matlab R2022a and is available at https://github.com/pasplin/symptom-propagation-mathematical-modelling.

To explore the effect of symptom propagation on the epidemiological dynamics of our three exemplar pathogens, for set values of $\alpha$ (0 through 1 in 0.1 increments) and a fixed value of $\nu$ (0.2), we calculated the following quantities: the final outbreak size (the number of recovered individuals at the end of the simulation), the peak prevalence (the maximum number of individuals infected at any one time) and the outbreak duration (the number of days until cumulatively there was less than one infected individual across all of the infected classes).

We also inspected the dependence of the proportion of infections that were severe with respect to changes in $\alpha$ and $\nu$ for the three exemplar pathogens. We tested combinations of $\alpha$-$\nu$ values with an increment of 0.05 for each parameter.

Across all simulations, for each value of $\alpha$ (and $\nu$), $\beta$ was chosen to generate a given $R_0$ of the required value for that disease parameterisation (further details in Section 1 in S1 Text).

## Modelling interventions

To investigate how the strength of symptom propagation could impact epidemiological and health economic assessments of infectious respiratory disease intervention strategies, we considered the roll-out of three vaccines with different mechanistic actions on the infectious disease dynamics: (a) a symptom-attenuating vaccine; (b) an infection-blocking vaccine (that did not impact the severity of any breakthrough infections); (c) an infection-blocking vaccine that only admitted mild breakthrough infections. Our vaccines having differing modes of action was motivated by contemporary studies on effectiveness of SARS-CoV-2 vaccines that have estimated effectiveness with respect to infection, symptoms, hospitalisation and mortality [14–16].

We assumed a proportion of the population, given by the vaccine uptake, $u$, were vaccinated before the start of the outbreak. In model terms, we moved a proportion, $u$, of those who would initially be in the susceptible ($S$) class to the vaccinated class, $V$.

We assumed that all three vaccines were imperfect (below 100% efficacy, with details on the assumed vaccine efficacies stated in the upcoming simulation overview subsection) and had a 'leaky' action, meaning the susceptibility of all those who were vaccinated was modulated by the vaccine efficacy- in comparison, an 'all-or-nothing' action would result in some of those who were vaccinated having full protection and the rest remaining fully susceptible. To minimise complexity, in each simulation, we only studied sole intervention use.

We highlight that the proportion of cases that were severe had a strong dependence on $\alpha$, meaning $\alpha$ would have a notable impact on the observed effectiveness of vaccination strategies, even when we would intuitively expect symptom propagation to have no effect. Consequently, we sought to separate the impact of symptom propagation on vaccination strategies from confounding epidemiological factors that would result in an increase in severe cases. Therefore, we explored the effectiveness of the above-described three vaccines for two values of $\alpha$, 0.2 and 0.8, where for each value of $\alpha$ we chose the appropriate value of $\nu$ to fix the proportion of cases (in the absence of interventions) that were severe at 80%. This value was chosen to allow a large value of $\alpha$ to be considered, as in this case, the proportion of cases that were severe was high regardless of the value of $\nu$. Conceptually, we may consider that the

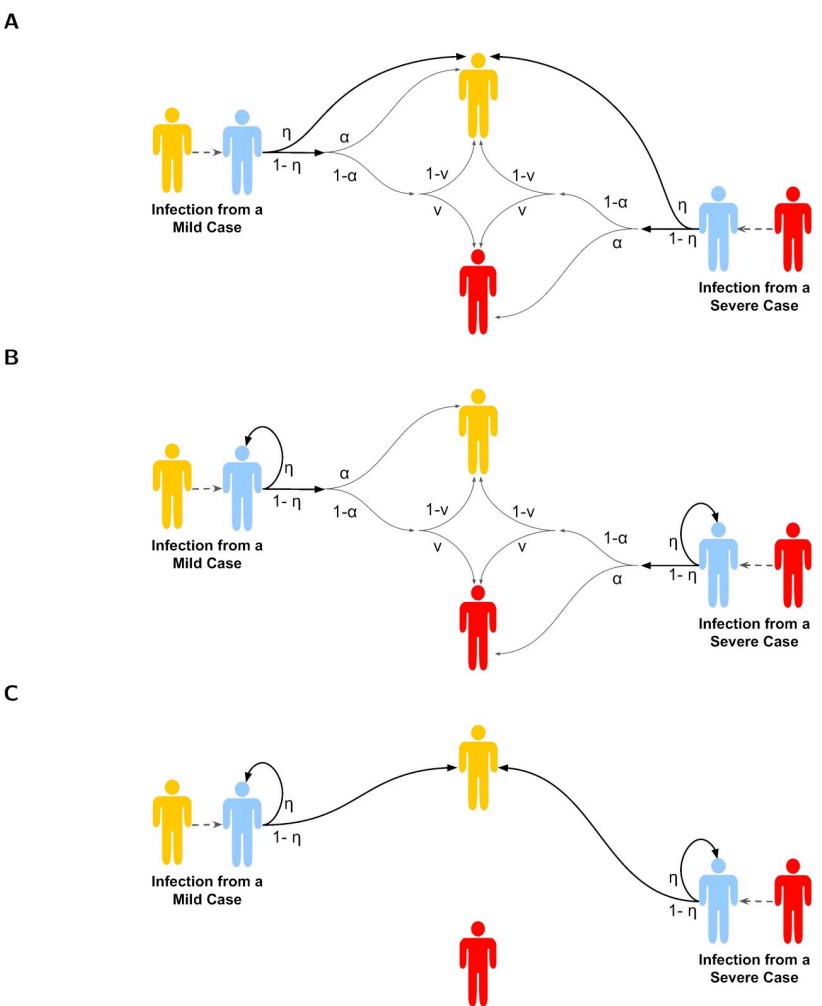

**Fig 2. Schematics depicting the mechanistic actions of each vaccine.** Across all panels, yellow shaded individuals correspond to infectious cases with mild symptoms, red shaded individuals correspond to infectious cases with severe symptoms, blue shaded individuals correspond to those who are vaccinated. The values on the arrows show the corresponding probability. The three vaccines displayed are: **(A)** symptom-attenuating vaccine—an infected individual who is vaccinated had a probability $\eta$ of having mild disease, and a probability $1 - \eta$ of their symptom severity being determined as usual; **(B)** an infection-blocking vaccine—a vaccinated individual had a probability $\eta$ of their infection being prevented, and a probability $1 - \eta$ of being infected and their symptom severity being determined as usual; and **(C)** an infection-blocking vaccine that only admits mild breakthrough infections—a vaccinated individual had a probability $\eta$ of their infection being prevented, and a probability $1 - \eta$ of being infected but only with mild disease.

proportion of severe cases is 'known' from epidemiological data, and we are fitting parameters to match this data.

We next provide an overview of the modifications to the model equations for each of the three vaccines (depicted in Fig 2). For each vaccine, we show the modified equations for $E_M$ and $E_S$ states, with the equations for all other states being unchanged from those in Eq 1. We also introduce a $V$ class, corresponding to susceptible individuals who are vaccinated.

**(a) Symptom-attenuating vaccine.** First, we considered a vaccine with a symptom-attenuating effect. Vaccinated individuals have probability $\eta$ of having mild disease, and probability $1 - \eta$ of their symptom severity being determined as usual (Fig 2A).

Updates to the $V$ class, corresponding to susceptible individuals who are vaccinated, and the exposed classes ($E_S$ and $E_M$) followed these ODEs:

$$\frac{dV}{dt} = -(\lambda_M + \lambda_S)V$$

$$\frac{dE_M}{dt} = \eta(\lambda_M + \lambda_S)V$$
$$+((\alpha + (1-\alpha)(1-v))\lambda_M + (1-\alpha)(1-v)\lambda_S)(S + (1-\eta)V) - \epsilon E_M$$

$$\frac{dE_S}{dt} = ((1-\alpha)v\lambda_M + (\alpha + (1-\alpha)v)\lambda_S)(S + (1-\eta)V) - \epsilon E_S.$$

(3)

**(b) Infection-blocking vaccine.**   Next, we considered a vaccine with an infection-blocking effect. Vaccinated individuals are not infected when exposed with probability $\eta$ (Fig 2B); otherwise, infection and symptoms proceed as before.

The revised system equations were:

$$\frac{dV}{dt} = -(1-\eta)(\lambda_M + \lambda_S)V$$

$$\frac{dE_M}{dt} = ((\alpha + (1-\alpha)(1-v))\lambda_M + (1-\alpha)(1-v)\lambda_S)(S + (1-\eta)V) - \epsilon E_M$$

$$\frac{dE_S}{dt} = ((1-\alpha)v\lambda_M + (\alpha + (1-\alpha)v)\lambda_S)(S + (1-\eta)V) - \epsilon E_S.$$

(4)

**(c) Infection-blocking vaccine that only admits mild breakthrough infections.**   Lastly, we modelled a vaccine with a combined infection-blocking and symptom-blocking effect. The action of this vaccine meant those who were protected were not infected when exposed with probability $\eta$. Furthermore, all vaccinated individuals only develop mild disease, regardless of the efficacy of the vaccine in blocking infection (Fig 2C).

The revised system equations were:

$$\frac{dV}{dt} = -(1-\eta)(\lambda_M + \lambda_S)V$$

$$\frac{dE_M}{dt} = (1-\eta)(\lambda_M + \lambda_S)V$$
$$+((\alpha + (1-\alpha)(1-v)\lambda_M + (1-\alpha)(1-v)\lambda_S)S - \epsilon E_M$$

$$\frac{dE_S}{dt} = ((1-\alpha)v\lambda_M + (\alpha + (1-\alpha)v)\lambda_S)S - \epsilon E_S.$$

(5)

Additionally, we modelled another infection-blocking vaccine that had an alternative manner in which breakthrough infections could arise, namely breakthrough infections were only possible when the infector was a severe case. We found no difference in results between this vaccine type and the infection-blocking vaccine. We provide the model schematic, equations and summary of findings in Section 4 in S1 Text.

**(d) Exploring the effect of symptom propagation on epidemiological dynamics in the presence of vaccination: Simulation overview.**   Across our main vaccination analyses, the vaccine efficacy was fixed at 70% based on estimates of COVID-19 vaccine efficacy [14, 15].

This vaccine efficacy level was also a reasonable selection for influenza vaccines since the vaccine efficacy during the 2009/10 season was estimated to be 72% vs the pandemic H1N1 strain [63], and other studies have estimated influenza vaccine efficacy to be between 56–78% [64] and between 26–73% [65]. To assess sensitivity to vaccine efficacy, as supplementary analyses, we also considered vaccine efficacies of 50% and 90%, where we found qualitatively similar results (see Section 7 in S1 Text, Figs P-W).

For the three vaccine types, two vaccine uptake levels (50% and 90%) and three sets of disease parameters, we calculated the proportion of the population in each recovered compartment (having had mild or severe symptoms) at the end of the outbreak for two values of $\alpha$ (0.2 and 0.8), with $\nu$ chosen to fix the proportion of cases that were severe without intervention at 80%.

For a range of values of $\alpha$ and $\nu$ (0 through 1 in 0.02 increments), we also calculated the difference in the number of individuals severely infected when an infection-blocking vaccine was used compared to a symptom-attenuating vaccine.

## Health economic modelling

Often there are many potential intervention strategies that can be used to limit the spread of the disease. Since public health decision makers have a finite budget, the cost of the intervention is an important factor to consider alongside the resulting epidemiological outcomes. Thus, measures for both the benefit to public health and the costs associated with the intervention and treatment warrant consideration.

**Measures of health quality and model parameterisation.**   The chosen measure to quantify disease burden was quality-adjusted life years (QALYs), which consider both the quality and quantity of years lived [66]. We assumed there were no QALY losses associated with mild cases. The magnitude of QALY loss from a severe case depended on whether the case was hospitalised and whether it was fatal; a proportion of severe cases was assumed to lead to hospitalisation and fatalities, as dictated by the pathogen-specific hospitalisation rate and death rate (parameter details in Table 2). Hospitalisations and fatalities also had an associated monetary cost, where once again values differed for influenza and SARS-CoV-2 (Table 2). Further details of the health economic model parameters are provided in Section 3 in S1 Text.

We deemed an intervention to be cost-effective if the overall cost of implementing the intervention was less than or equal to the value of QALYs gained from doing so. In particular, we computed threshold unit intervention costs, the monetary cost of an intervention unit that would result in intervention costs equalling the monetary value of QALYs gained. In this case, the threshold unit intervention cost refers to the threshold cost per vaccine dose.

Determining whether an intervention is cost-effective requires setting a willingness to pay (WTP) threshold per QALY—the amount one is willing to pay to gain one QALY. We used a default WTP per QALY of £20,000, reflecting the typical criteria used in England that alternative intervention strategies need to satisfy to be judged as cost-effective [71]. Equivalently,

$$\text{Threshold unit intervention cost} = \frac{(\text{WTP threshold}) \times (\text{QALY loss prevented}) + (\text{hospital costs prevented})}{\text{intervention uptake} \times N}. \tag{6}$$

In all simulations, the threshold unit intervention cost was normalised with respect to the highest absolute threshold unit intervention cost attained for that disease parameterisation across the range of tested vaccine uptake values.

**Exploring the effect of symptom propagation on health economic outcomes: Simulation overview.**   For the three vaccine types, two vaccine uptake levels (50% and 90%) and three

**Table 2. Description of parameters used in the health economic modelling. (Top)** Values applied to both the seasonal influenza and pandemic influenza disease parameterisations. **(Bottom)** Values applied to the SARS-CoV-2 disease parameterisation.

| Description | Influenza value | Source |
|---|---|---|
| Probability of hospitalisation (given severe disease) | 0.01 | Hill et al. [42] |
| Probability of death (given severe disease) | 0.001 | Hill et al. [42] |
| QALY loss from a mild case | 0 QALYs | Assumption |
| QALY loss from a severe, non-hospitalised case | 0.008 QALYs | Hill et al. [42] |
| QALY loss from a non-fatal hospitalised case | 0.018 QALYs | Hill et al. [42] |
| QALY loss from a fatal hospitalised case | 37.5 QALYs | Hollmann et al. [67] |
| Total cost of a non-fatal hospitalised case | £1,300 | Hill et al. [42] |
| Total cost of a fatal hospitalised case | £2,600 | Hill et al. [42] |
| Willingness to pay threshold per QALY | £20,000 | NICE [68] |

| Description | SARS-CoV-2 value | Source |
|---|---|---|
| Probability of hospitalisation (given severe disease) | 0.065 | Moran et al. [69] |
| Probability of death (given severe disease) | 0.02 | Moran et al. [69] |
| QALY loss from a mild case | 0 QALYs | Assumption |
| QALY loss from a severe, non-hospitalised case | 0.0035 QALYs | Moran et al. [69] |
| QALY loss from a non-fatal hospitalised case | 0.0059 QALYs | Moran et al. [69] |
| QALY loss from a fatal hospitalised case | 11.29 QALYs | Moran et al. [69] |
| Total cost of a non-fatal hospitalised case | £2,600* | Vekaria et al. [70] |
| Total cost of a fatal hospitalised case | £5,200* | Vekaria et al. [70] |
| Willingness to pay threshold per QALY | £20,000 | NICE [68] |

* Hospitalisation costs for SARS-CoV-2 are double those of influenza, based on the average hospital stay being twice as long for SARS-CoV-2 versus influenza—further details available in Section 3 in S1 Text.

disease parameterisations, we conducted comparisons of the relative threshold unit intervention cost between two values of $\alpha$ (0.2 and 0.8) with $v$ chosen to fix the proportion of cases that were severe without intervention at 80%. We then explored how the threshold unit intervention costs varied with the vaccine uptake, looking at uptake levels between 0% and 100% in 1% increments.

For the results presented in the main text, we applied a 3.5% discounting rate to both QALY losses and monetary costs, as recommended [72]. For sensitivity purposes, we also tested having no discounting, with results given in Section 6 in S1 Text (Figs N and O).

## Results

### Symptom propagation affects epidemiological dynamics

Initially, we explored the effect of varying the strength of symptom propagation on epidemiological outcomes for a fixed baseline probability of severe disease, $v = 0.2$. Since, for each set of parameters, the value of $R_0$ was fixed, the resultant outbreak size remained mostly constant as we varied the strength of symptom propagation, $\alpha$ (Fig 3A–3C), although there was a slight reduction in final outbreak size as $\alpha$ approached 0.5 due to numerical inaccuracies (Fig F in S1 Text). As these differences in case numbers were proportionally small, we assumed the final size to be fixed throughout the remainder of the analysis.

Considering the stratification of cases by severity (mild or severe), the proportion of total cases that were severe monotonically increased with $\alpha$. As expected, when $\alpha = 0$, the

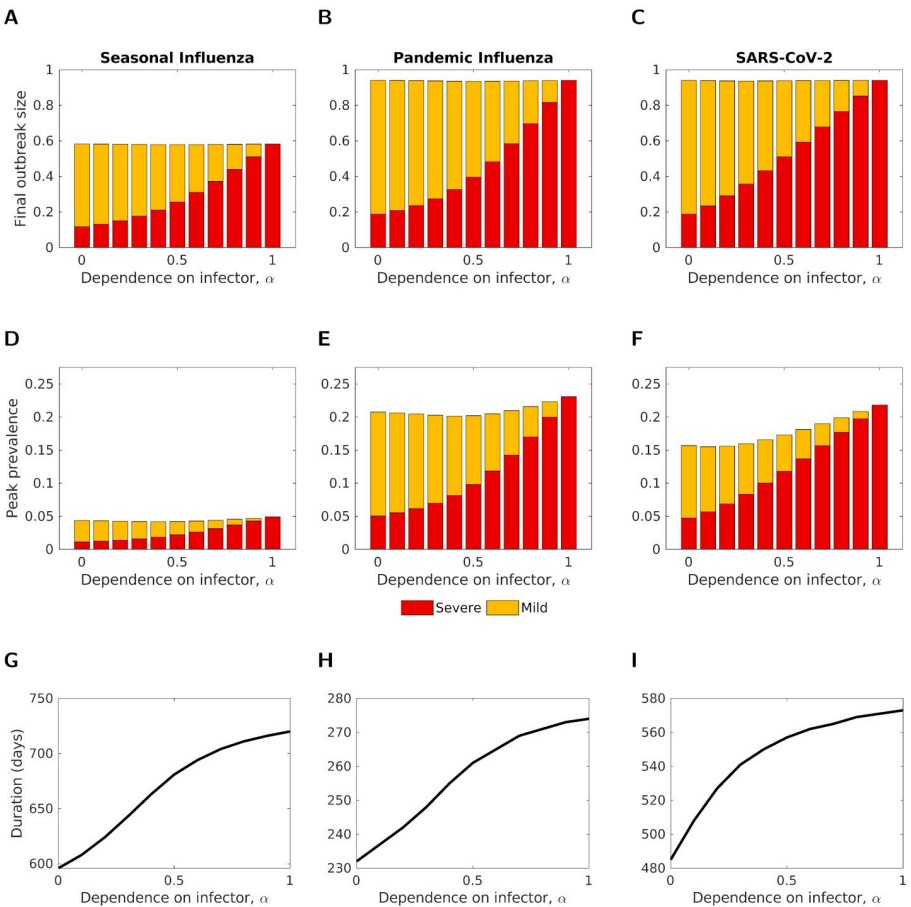

**Fig 3. The final outbreak size, peak prevalence and outbreak duration, by severity, for three disease parameterisations, plotted for different symptom propagation strengths, $\alpha$. (A-C)** Final outbreak size by severity. **(D-F)** Peak prevalence by severity. The intensity of the shading denotes the symptom severity class, with severe cases in red and mild cases in yellow. **(G-I)** Outbreak duration (note the different y-axis scales). In all panels, $v = 0.2$. The three disease parameterisations used were: **(A,D,G)** Seasonal influenza; **(B,E,H)** Pandemic influenza; **(C,F,I)** SARS-CoV-2, with parameters as given in Table 1.

proportion of cases that were severe was equal to the baseline probability of severe disease, $v = 0.2$. This proportion increased to effectively all cases being severe when $\alpha = 1$ (Fig 3A–3C). Similarly, the proportion of cases that were severe at the peak of the outbreak increased with $\alpha$, with this proportion aligning with the proportion severe overall (Fig 3D–3F). The outbreak duration also increased with $\alpha$ (Fig 3G–3I) due to individuals with severe disease having a longer infectious duration. Comparing $\alpha = 1$ with $\alpha = 0$ across all three disease scenarios, we observed an increase of approximately 20% in the outbreak duration.

Overall, the qualitative patterns of the impact of symptom propagation differed relatively little between the three sets of disease parameters. Unsurprisingly, seasonal influenza had a much lower final outbreak size and peak prevalence than the other two disease parameterisations (Fig 3A and 3D, due to its lower assumed $R_0$ value). More interestingly, SARS-CoV-2 had a lower peak prevalence than pandemic influenza, especially for low values of $\alpha$ (Fig 3E and 3F), with longer generation times for SARS-CoV-2 dominating the effects of a higher $R_0$ value. SARS-CoV-2, compared to the two influenza disease parameterisations, showed a slightly higher proportion of severe cases for intermediate values of $\alpha$. Furthermore, the

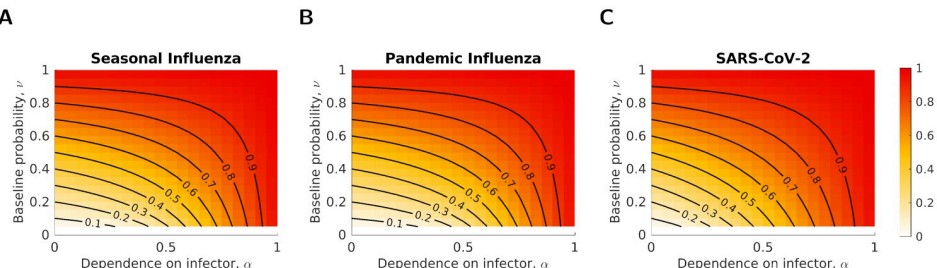

**Fig 4. The proportion of infections that were severe against changes in $\alpha$ and $v$ for three parameter sets.** The shading shows the proportion of infections that were severe, with darker shading corresponding to a higher proportion being severe. Black lines correspond to the contours taking values from 0.1 to 0.9, at increments of 0.1. The three disease parameterisations used were: **(A)** Seasonal influenza; **(B)** Pandemic influenza; **(C)** SARS-CoV-2. The parameters were as given in Table 1.

proportion of cases that were severe (both overall and at peak) for SARS-CoV-2 exhibited a roughly linear increase as $\alpha$ increased from 0 to 1, whereas for influenza these metrics increased sub-linearly for $\alpha$ between 0 and 0.5, and approximately linearly for increasing $\alpha$ between 0.5 and 1 (Fig 3A–3C).

Next, we considered how the value of the baseline probability of severe disease, $v$, impacted epidemiological outcomes. As expected, the proportion of cases that were severe increased with $v$ (Fig 4). For values of $\alpha$ close to 0, the value of $v$ mostly determined the proportion of cases that were severe. In contrast, when $\alpha$ was close to 1, the proportion of cases that were severe remained high, independent of the value of $v$. The relationship between $\alpha$, $v$ and the proportion of cases that were severe was consistent across parameter sets.

## Symptom propagation increases the effectiveness of interventions that impact symptom severity

In this section, we explored three types of intervention, corresponding to three plausible vaccination scenarios: a symptom-attenuating vaccine (SA), an infection-blocking vaccine (IB) and an infection-blocking vaccine that only admits mild breakthrough infections (IB_MB). We considered two vaccine uptake rates (50% and 90%) and two values of $\alpha$ (0.2 and 0.8). In order to highlight the differences between vaccine types, the value of $v$ was chosen (as a function of a given value of $\alpha$) to fix the proportion of cases that were severe equal to 0.8. We additionally produced an analogous set of results with $v$ fixed equal to 0.2 for comparison (see Section 5.4 in S1 Text, Figs L and M), noting there would be an inherent relative reduction in potential impact of SA interventions for that scenario.

For all parameter sets and both uptake and $\alpha$ values, the IB_MB vaccine type was, unsurprisingly, the most effective at reducing both total and severe cases (Fig 5). Similarly, a solely infection-blocking vaccine was always more effective at reducing total cases than a symptom-attenuating vaccine. In the case of seasonal influenza, both IB and IB_MB were sufficient to fully suppress the outbreak (<0.01% of the population was infected), even at 50% uptake, whereas 90% uptake was required for the SA vaccine to suppress the outbreak. However, in many cases, the symptom-attenuating vaccine was more effective at reducing severe cases than the solely infection-blocking vaccine. This effect was seen for both $\alpha$ values for pandemic influenza or SARS-CoV-2 and 50% uptake (Fig 5B and 5C). For these two disease parameterisations, when the uptake was instead 90%, whether the SA or IB vaccine was more effective at reducing severe cases depended on the value of $\alpha$, with the higher value of $\alpha = 0.8$ resulting in the SA vaccine being more effective (Fig 5E and 5F).

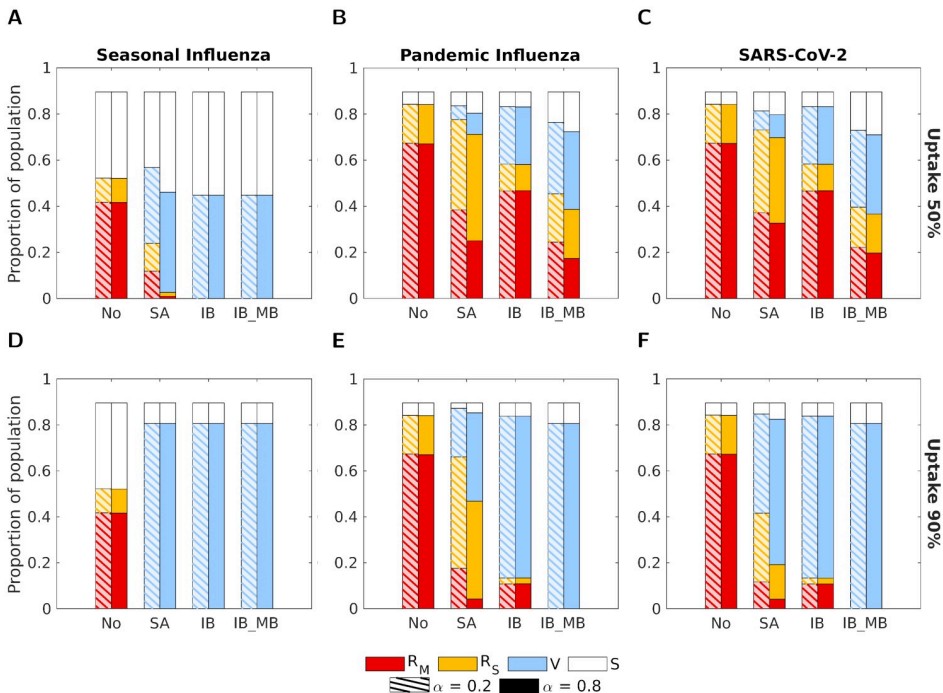

**Fig 5. The proportion of the population in each disease state at the end of the outbreak for the four intervention scenarios, two vaccine uptake levels and three disease parameterisations.** The four groups of bars correspond to four intervention scenarios: no intervention (No), a symptom-attenuating vaccine (SA), an infection-blocking vaccine (IB) and an infection-blocking vaccine which only admits mild breakthrough infections (IB_MB). The two bars in each group correspond to two different strengths of symptom propagation: $\alpha = 0.2$ (left bar with hatched lines) and $\alpha = 0.8$ (right bar with solid fill). Bar shading corresponds to the disease status: red—recovered from severe infection ($R_S$); yellow—recovered from mild infection ($R_M$); blue—susceptible and vaccinated ($V$); white—susceptible and not vaccinated ($S$). The two rows correspond to two vaccine uptake levels: **(A-C)** 50%; **(D-F)** 90%. Columns correspond to different disease parameterisations: **(A,D)** seasonal influenza; **(B,E)** pandemic influenza; **(C,F)** SARS-CoV-2. We fixed the vaccine efficacy at 70% and all other parameters as given in Table 1, with $\nu$ chosen to fix the proportion of cases that were severe equal to 0.8.

Across all scenarios, the IB vaccine resulted in the same epidemiological outcomes for both values of $\alpha$. For the other two vaccine types (SA and IB_MB), effectiveness was always higher for the higher $\alpha$ value, with the exception of scenarios in which the outbreak was suppressed for both $\alpha$ values (<0.01% of the population was infected). Results were similar when considering peak prevalence (Fig J in S1 Text). For all intervention scenarios (including no intervention), the duration was higher for $\alpha = 0.8$ compared to $\alpha = 0.2$ for those instances where the outbreak was not effectively prevented (>0.01% of the population was infected, Fig K in S1 Text). Inspection of temporal profiles of the outbreaks revealed that, in all intervention scenarios, the peak prevalence occurred later for $\alpha = 0.8$ than for $\alpha = 0.2$, even when no intervention was used (Figs G-I in S1 Text). For the no intervention and IB vaccine scenarios, this delay was a noticeable change in the temporal dynamics between the two $\alpha$ values; otherwise, the temporal dynamics largely exhibited similar qualitative behaviour.

We then explored the difference in the number of severe cases prevented by an IB vaccine and an SA vaccine for the three disease parameterisations and two vaccination uptake levels, under a fixed vaccine efficacy (70%) (Fig 6). We found that the IB vaccine was always more effective at preventing severe cases in the case of seasonal influenza (Fig 6A and 6D). For the other two parameter sets, the results were qualitatively similar. For a lower vaccine uptake

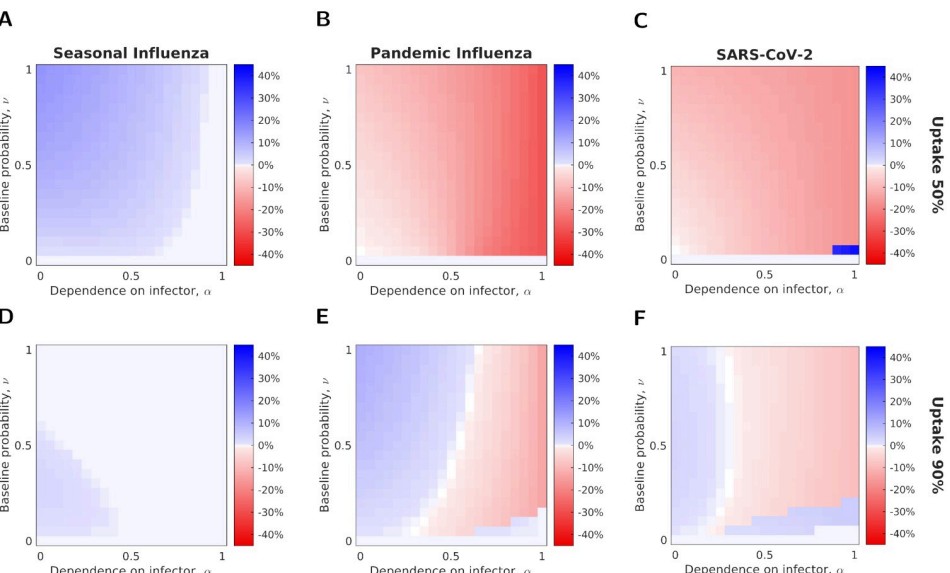

**Fig 6. The relative effectiveness in reducing severe cases of a symptom-attenuating and infection-blocking vaccine as a function of $\alpha$ and $v$, given a fixed efficacy (70%).** Rows correspond to the two vaccine uptake levels **(A-C)** 50%; **(D-F)** 90%, and columns to the different disease parameterisations (**(A,D)** seasonal influenza; **(B,E)** pandemic influenza; **(C,F)** SARS-CoV-2). Pixel shading denotes (for given combinations of $\alpha$-$v$ values) the difference in the proportion of the population severely infected between vaccine types, such that blue regions show parameter combinations where the infection-blocking vaccine was more effective at reducing the number of severe cases and red regions show those where the symptom-attenuating vaccine was more effective.

(50%), the SA vaccine was more effective at reducing severe cases for almost all values of $\alpha$ and $v$ (red shaded cells of Fig 6B and 6C). In contrast, when the vaccine uptake was high (90%), which vaccine type was most effective at reducing severe cases depended on the values of $\alpha$ and $v$, with the SA vaccine being more effective for larger values of $\alpha$ and $v$ (Fig 6E and 6F).

## Symptom propagation can affect health economic outcomes

Switching attention to how the strength of symptom severity propagation can impact health economic assessments, the differences in the threshold unit intervention cost for the two values of $\alpha$ aligned with our previously presented results (Fig 7).

When comparing between vaccine types, we found that the IB_MB vaccine always had the highest threshold unit intervention cost of the three vaccine types considered. For the pandemic influenza and SARS-CoV-2 parameterisations, SA always had a higher threshold unit intervention cost (and was, therefore, more cost-effective) than IB when $\alpha = 0.8$ (Fig 7B, 7C, 7E and 7F). When $\alpha = 0.2$, SA was only more cost-effective than IB when uptake was low (50%) (Fig 7B and 7C).

When comparing between the two $\alpha$ values, there was the least variation observed for the IB vaccine, where the threshold unit intervention cost was roughly the same for the two $\alpha$ values in all uptake and disease parameterisation scenarios. This result aligns with our previous finding that $\alpha$ had no effect on epidemiological outcomes when an IB vaccine was used (Fig 5). For the other two vaccine types, SA and IB_MB, the threshold unit intervention cost was higher when $\alpha = 0.8$, unless the outbreak was effectively contained (<0.01% of the population was infected) for both $\alpha$ values, in which case the threshold unit intervention costs were the same. In all of these scenarios, the relative difference in threshold unit intervention cost was

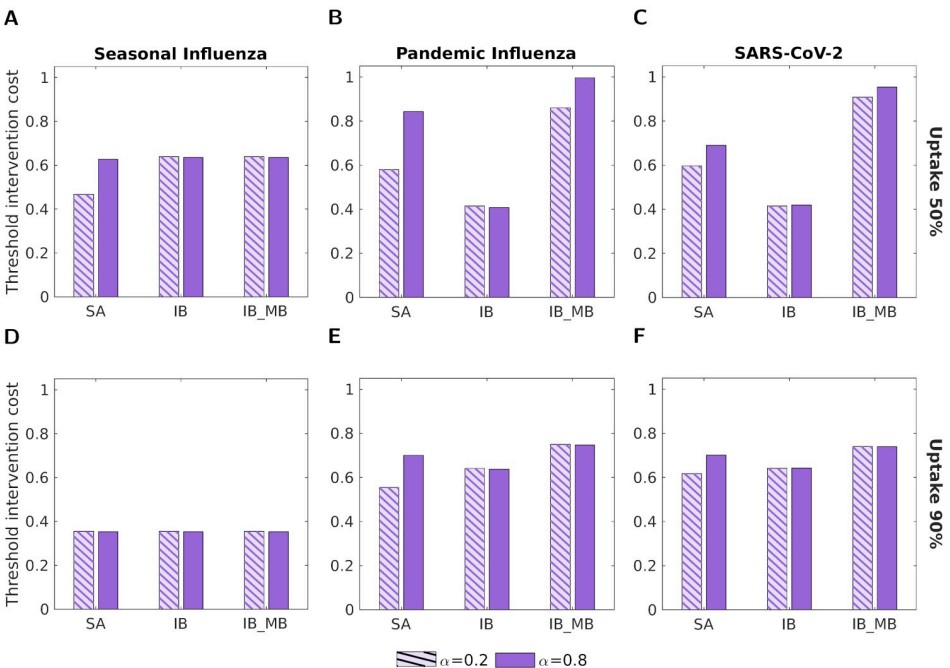

**Fig 7. Threshold unit intervention cost for the three types of vaccine, two vaccine uptake levels and the three disease parameterisations.** In all panels, we normalised threshold intervention costs by the highest absolute threshold unit intervention cost obtained across the range of tested vaccine uptake values. The three groups of bars correspond to: a symptom-attenuating vaccine (SA), an infection-blocking vaccine (IB) and an infection-blocking vaccine which only admits mild breakthrough infections (IB_MB). The two bars in each group correspond to symptom propagation strengths of $\alpha = 0.2$ (left bar, hatched lines) and $\alpha = 0.8$ (right bar, solid fill). The two rows correspond to two vaccine uptake levels: **(A-C)** 50%; **(D-F)** 90%. Columns correspond to differing disease parameterisations: **(A,D)** seasonal influenza; **(B,E)** pandemic influenza; **(C,F)** SARS-CoV-2. Vaccine efficacy was fixed at 70% and all other parameters were as given in Table 1, with $\nu$ chosen to fix the proportion of cases that were severe equal to 0.8.

higher for the influenza parameter sets (16%-45% increase, Fig 7A, 7B, 7D and 7E) than for the SARS-CoV-2 parameter set (Fig 7C and 7F, 5%-16% increase). The relative difference in outcomes between $\alpha$ values was also generally higher for the SA vaccine than for the IB_MB vaccine (14%-45% increase for SA vs 5%-16% for IB_MB). In all cases, as anticipated the observed differences in threshold unit intervention cost reflect the differences in epidemiological outcomes (Fig 5).

We then explored how vaccine cost-effectiveness varied for a range of $\alpha$ and $\nu$ values (see Section 8.1 in S1 Text, Figs X-Z). We found that the threshold unit intervention cost increased with both $\alpha$ and $\nu$ irrespective of the action of the vaccine as a consequence of higher values of $\alpha$ and $\nu$ causing a larger proportion of cases to be severe.

We then applied greater scrutiny to how the threshold unit intervention cost depended on the level of uptake for the three vaccine types and three disease parameterisations (Fig 8). For both seasonal influenza (Fig 8A–8C) and pandemic influenza (Fig 8D–8F), the difference in threshold unit intervention cost (obtained for $\alpha = 0.2$ and $\alpha = 0.8$) decreased as vaccine uptake increased. This was particularly noticeable for the SA vaccine, where the difference decreased somewhat linearly until the point where the values converged. For the SARS-CoV-2 parameterisation, the difference in threshold unit intervention cost between the two $\alpha$ values remained relatively constant as the uptake increased, up until the point where the two values converged (Fig 8G–8I).

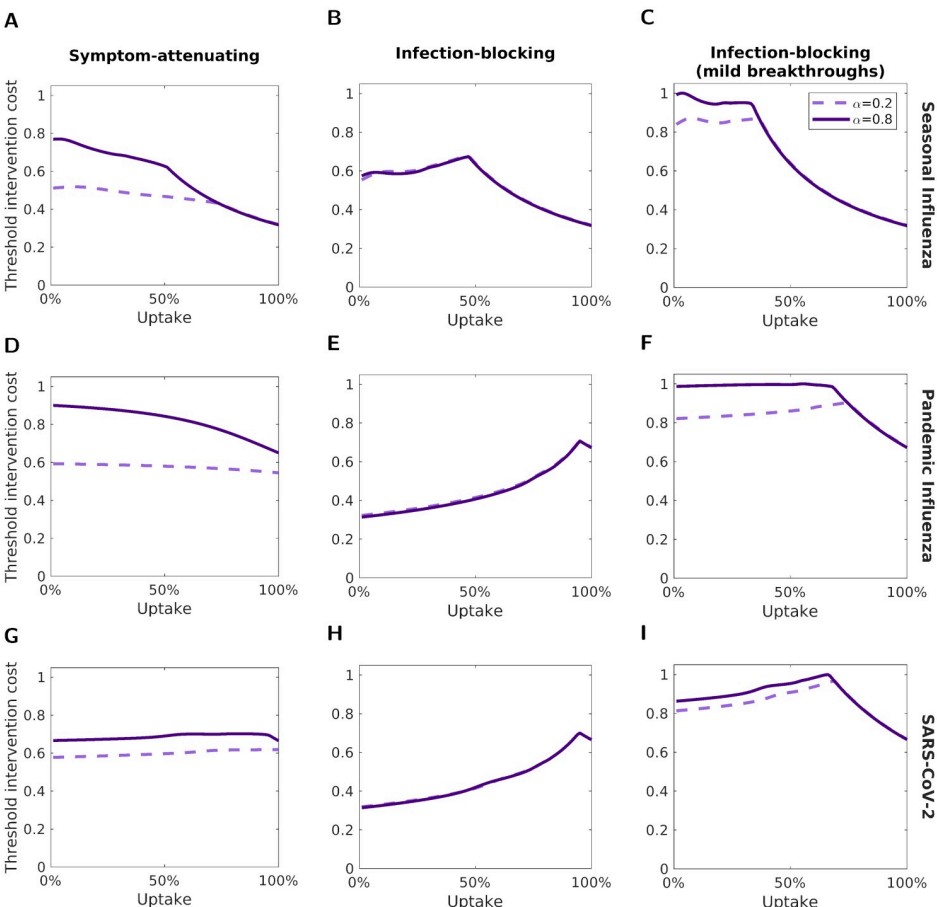

**Fig 8. Variation of the threshold unit intervention cost with vaccine uptake.** We normalised threshold unit intervention costs for each disease parameterisation; the normalisation constant was the highest absolute threshold unit intervention cost attained for the respective disease parameterisation across the range of tested uptake values. The three rows correspond to the three different disease parameterisations: **(A-C)** seasonal influenza; **(D-F)** pandemic influenza; **(G-I)** SARS-CoV-2. The three columns correspond to: **(A,D,G)** a symptom-attenuating vaccine (SA), **(B,E, H)** an infection-blocking vaccine (IB) and **(C,F,I)** an infection-blocking vaccine that only admits mild breakthrough infections (IB_MB). The two lines correspond to two symptom propagation strengths; the dashed, light purple line corresponds to $\alpha = 0.2$ and the solid, dark purple line corresponds to $\alpha = 0.8$. We fixed the vaccine efficacy at 70% and all other parameters values were as given in Table 1, with $v$ chosen to fix the proportion of cases that were severe equal to 0.8.

As previously, we observed different effects of $\alpha$ for the different types of vaccine. For the IB vaccine, we found that the threshold unit intervention cost was equal for the two values of $\alpha$ at all vaccine uptake levels (Fig 8B, 8E and 8H). For the SA vaccine (Fig 8A, 8D and 8G) and IB_MB vaccine (Fig 8C, 8F and 8I), the threshold unit intervention cost remained higher for $\alpha$ = 0.8 as the uptake increased, up until the uptake level at which the outbreak was suppressed for both $\alpha$ values. After this point, for both $\alpha$ values, the threshold unit intervention costs were equal and decreased monotonically with vaccine uptake.

Across all uptake and disease parameterisation scenarios, the most cost-effective uptake (i.e. the uptake that maximised the threshold unit intervention cost) varied minimally with $\alpha$. There was a consistent general relationship between the threshold unit intervention cost and the vaccine uptake between the two $\alpha$ values.

We further explored the most cost-effective uptake value for a range of $\alpha$ and $v$ values and found that the most cost-effective level of vaccine uptake (i.e the uptake with the highest threshold unit intervention cost) generally remained constant across $\alpha$ and $v$ values (see Section 8.2 in S1 Text, Figs AA-AC). Some exceptions to this did arise for the vaccines that were symptom-attenuating and infection-blocking with mild breakthrough infections. For certain disease parameterisations and efficacy values, we found large variation in the most cost-effective uptake, from close to 0% to nearly 100% in some parameterisations. Between scenarios, we did not observe a simple qualitative pattern of variation, although the regions where uptake close to 0% was most cost-effective tended to have lower values of $\alpha$ and $v$.

Lastly, to give insight into how outcomes could differ if symptom propagation was mistakenly omitted from the modelled infection dynamics, we compared between the threshold unit intervention cost of the most cost-effective uptake for a particular value of $\alpha$ and the threshold unit intervention cost for the most cost-effective uptake at $\alpha = 0$ (see Section 8.3 in S1 Text, Figs AD-AF). We found that generally the difference in threshold unit intervention costs increased with $\alpha$ for both the SA vaccine and the IB_MB vaccines.

## Discussion

In this paper, we make three main contributions to the literature. Firstly, we introduce a parsimonious and generalisable mechanistic mathematical framework to model infectious disease transmission that incorporates symptom propagation of different strengths via a single parameter, $\alpha$. Secondly, we demonstrate substantial impacts of symptom propagation on epidemiological outcomes. For parameterisations corresponding to seasonal influenza, pandemic influenza and SARS-CoV-2, we demonstrate that, for a given value of $R_0$, even for a low baseline probability of severe disease, $v$, (which we conceptualise as relating to the virulence of the pathogen) the proportion of cases that experience severe disease can approach one as the strength of symptom propagation, $\alpha$, increases. Thirdly, we apply three types of intervention, corresponding to three plausible vaccination scenarios (symptom-attenuating, SA, infection-blocking, IB, and infection-blocking with mild breakthrough infections, IB_MB), and demonstrate important impacts of symptom propagation on epidemiological and health economic outcomes. We showed that although the strength of symptom propagation had no effect on epidemiological outcomes for IB interventions, differences were seen for interventions that acted to reduce symptom severity, with the effectiveness of these interventions in reducing the number of severe and total cases increasing with the strength of symptom propagation. The strength of symptom propagation also affected the relative effectiveness of SA and IB interventions in reducing severe cases, with the optimal type of intervention dependent on a combination of uptake and $\alpha$. In the health economic analysis, we found that the strength of symptom propagation had important implications for the cost-effectiveness of SA interventions and, to a lesser extent, IB_MB interventions. Thus, we demonstrated how symptom propagation influences both epidemiological and health economic outcomes, which can alter the balance between preferred intervention types.

A cornerstone of our encapsulation of symptom propagation was the parameter $\alpha$, the dependence of the symptoms of an infected individual on the symptoms of their infector, or equivalently, the strength of symptom propagation. Our general finding was that the proportion of cases that were severe increased with $\alpha$. Although the effect of symptom propagation on the proportion of cases that are severe has not received attention in previous modelling studies, this result aligns with suggestions that symptom propagation, at least through a dose-response relationship, could lead to severe outbreaks and intense epidemics [25]. This result also aligns with the findings of Paulo et al. [36], who found that the inclusion of a dose-

response relationship in their model led to an increase in the incidence of severe disease and higher mortality. The appreciable effect of $\alpha$ on epidemiological outcomes motivates the inclusion of symptom propagation in models of infectious disease transmission, both when simulating an outbreak from a given set of parameters and when estimating parameters from an empirical data set. It also highlights the importance of examining in more detail the basic biological mechanisms of symptom propagation for respiratory pathogens.

We found that the proportion of cases that were severe not only increased with $\alpha$, but also with the baseline probability of severe disease, $v$. In many scenarios, for a given value of $\alpha$, it was possible to compute the value of $v$ that returned a pre-specified proportion of cases that were severe. However, this was limited to relatively low values of $\alpha$, since for high values of $\alpha$ the proportion of cases that were severe was large regardless of the value of $v$. As a result of this, when the proportion of cases that were severe was fixed, the proportion chosen was relatively high (80%) to allow for the consideration of a value of $\alpha$ close to one. We acknowledge that it may be unrealistic for such a high percentage of cases to have severe symptoms, with the proportion of cases that are mild estimated to be 43% for influenza (where mild is defined as subclinical) [73] and 44% for SARS-CoV-2 (where mild is defined as asymptomatic) [74]. It may also be the case that such high values of $\alpha$ (i.e. high amounts of symptom propagation) are unrealistic, although we know that for certain diseases, such as plague, the value of $\alpha$ is close to one [22]. Instead, this overestimation of the proportion of cases that are severe for high values of $\alpha$ may be due to our subjective parameterisation of some of the epidemiological parameters, for example, the relative transmissibility of mild and severe disease. Alternatively, it may be that such strengths of symptom propagation are realistic, but are not observed due to behavioural changes not included in this model, since individuals displaying symptoms have been shown to reduce their contact with others in their community [75]. The inclusion of behavioural responses is an important area of infectious disease modelling identified for dedicated research.

When exploring the effect of symptom propagation on the effectiveness of interventions, we found that interventions affecting symptom severity (i.e. the symptom-attenuating intervention or infection-blocking intervention with mild breakthroughs infections) were consistently more effective at reducing cases and were more cost-effective for a higher value of $\alpha$; exceptions to this were when the intervention suppressed the outbreak for both $\alpha$ values under consideration ($\alpha = 0.2$ and $\alpha = 0.8$). In contrast, varying $\alpha$ had little or no effect on epidemiological outcomes when the intervention was purely infection-blocking. These results suggest that determining the effect, if any, of an intervention on the symptoms experienced by individuals is critical to understand whether symptom propagation is an important consideration when investigating the effectiveness of the intervention.

Given increasing evidence that many interventions used to prevent the spread of disease also reduce symptom severity, it may be the case that symptom propagation should be considered more often than not. Indeed, vaccinated individuals are often more likely to have asymptomatic disease in addition to having a lower risk of infection, as was the case for COVID-19 vaccines [76]. Additionally, it has been indicated that non-pharmaceutical interventions such as social distancing and mask-wearing can reduce the infectious dose for onward transmission, leading to less severe disease in those infected [20, 21, 77].

For disease parameterisations with a higher $R_0$ (corresponding to pandemic influenza and SARS-CoV-2) and a high intervention uptake, comparisons between intervention types showed that, given a strong symptom propagation action, a symptom-attenuating intervention was more effective at reducing severe cases than an infection-blocking intervention, even though total outbreak sizes were larger. This finding suggests dual benefits of SA interventions: reduced numbers of severe cases alongside widespread population immunity. Similar effects

have been described previously. For example, a so-called 'variolation effect' has been discussed in the context of mask-wearing during the COVID-19 pandemic; some authors hypothesise that masks could act to reduce the inoculum dose leading to reduced disease severity in those infected, and that this effect could have been used to generate widespread immunity before vaccines became available [19, 21].

The intervention type ranked as most effective at reducing severe cases (when comparing between them) had a notable dependence on $\alpha$, $v$ and the disease parameterisation. Accordingly, symptom propagation may be an important factor to consider when choosing between intervention types. A renewed analysis of interventions for the containment, suppression and management of respiratory pathogens of public health concern could result, for example via the examination of interventions that can create large-scale population immunity while minimising the number of severe cases. Such modelling analysis is only viable by us taking a contemporary approach to capturing actions of interventions, rather than treating them as being solely infection blocking. The data arising from SARS-CoV-2 vaccines having a stratification of efficacy for different health episode outcomes (infection, symptomatic, hospitalisation, mortality) [14–16] motivates a similar richness of data collection being undertaken for other pathogens. Seasonal influenza is one such pathogen where additional information would be informative; vaccination effectiveness has historically been assessed using a 'test-negative' design, meaning patients with influenza-like illness are tested for influenza, with reported vaccine effectiveness usually relating solely to the prevention of symptomatic infection [78], without further stratification of outcomes.

There are a number of limitations to the work conducted in this paper. First, there was some uncertainty in the parameters used in the epidemiological model. We sourced parameters from the literature, where disparate estimates were reported. Certain parameters, such as the relative transmissibility of mild and severe disease, were difficult to measure. Whilst acknowledging that our results may be sensitive to the parameterisations chosen, to retain our focus on the implications of symptom propagation and interventions that had different modes of action, we took a pragmatic approach of considering a single fixed relative transmissibility scaling for each pathogen informed by the available literature. These relative transmissibility scalings were also dissimilar, two for influenza and four for SARS-CoV-2 giving us more breadth in our coverage of disease parameter space. A notable limitation of these choices is that the transmission rates do not account for changes in contact patterns that we would expect to see for those with more severe disease, and hence may be an overestimation of the real values. The inclusion of heterogeneity in contact patterns as a function of symptom severity is highlighted as an area for future work.

Additionally, whilst our intent is for our model to be generalisable to other respiratory pathogens for which symptom propagation is possible, the assumptions made in this paper may not be well suited to all such pathogens. In particular, the assumption that severe disease is more transmissible and has a longer infectious period may not hold. If this assumption were removed and mild and severe disease were to produce the same number of secondary cases, we would expect the proportion of cases that were severe to be similar regardless of the strength of symptom propagation.

Second, definitions of severity of infection can vary appreciably, showcased by prior work on the two pathogens focused upon in this study (influenza and SARS-CoV-2) [35, 41, 50, 51]. An inevitable consequence was there being uncertainty in quantifying health economic parameters (such as QALY losses and hospitalisation rates) for our 'mild' and 'severe' infection categorisations. Creating a formal definition of severity in this context requires a deeper biological understanding of how symptom propagation occurs and, in turn, further research. We would encourage a conceptual re-analysis of the symptoms of respiratory infections from a clinical

standpoint, leading to a new framework for categorising clinical outcomes informed by an understanding not only of patient-level symptoms, but also symptom propagation and its implications for onward transmission, along with the formulation of associated data collection protocols. Indeed, although we have chosen to focus on only two severity classes, this simplification may not always be appropriate. An extension of this model to include a separate asymptomatic class or a continuum of severity could be explored in future work (or even qualitatively different symptom sets that do not straightforwardly map to levels of severity), with model structural choices informed by data where appropriate.

We also recognise that there are many factors not included in this model which are known to affect symptom severity, and future work could extend the framework presented here to incorporate characteristics such as age, immune status and multiple strains. Age structure could be of particular interest because it has previously been suggested that the combination of age-dependent mixing and age-dependent severity might cause correlations between the severity of the infector and the infectee [41]. As such, the inclusion of age structure could amplify the effects of symptom propagation. Also, as a consequence of not modelling demographic characteristics, we assumed intervention uptake to be uniform across the population. In reality, those identified as risk groups, such as healthcare workers or immuno-compromised people, are likely to be targeted first as part of any intervention policy, as seen in the vaccine roll-out during the COVID-19 pandemic [79]. We expect that such targeting would amplify the increase in intervention effectiveness caused by strong symptom propagation and anticipate this effect could be seen for all intervention types, even purely infection-blocking interventions. Further work is required to investigate these potential dependencies.

In addition to extensions to the deterministic, compartmental ODE model used in this paper, the model framework could be applied to a range of model types [48]. If applied to a stochastic model at a localised spatial level (population size of the order of hundreds rather than millions), we could expect symptom propagation to result in a large variation in the proportion of cases that are severe, depending on the severity of the initial cases. We would likely find that, even for relatively weak symptom propagation, a stochastic model may generate large variation in the proportion of cases that are severe, with the potential for outbreaks to be predominantly severe. The symptom propagation model framework could also be applied to network or spatial models. In these cases, we might expect symptom propagation to result in large spatial heterogeneity in the severity of (local) outbreaks, leading to increased strain on local healthcare services despite the larger-scale outbreak severity being similar to what is predicted by a model with no spatial structure.

Another identified key area of future work is the estimation of the $\alpha$ parameter, i.e. the strength of symptom propagation. However, due to the complex nature of symptom severity and the many confounding factors, performing this inference is non-trivial. A large volume of individual data with both information on symptom severity and who infected whom is required. Major challenges include separating symptom propagation from the effects of strains and from the impact of genetic similarity between an individual and the person who infected them (e.g. in the case of related individuals). Nevertheless, the public health benefits of such estimations will make surmounting such challenges rewarding, including informing the relative importance of transmission from those who are asymptomatic (and therefore the optimal approach for contact tracing) and the role of vaccines that may reduce symptom severity as well as infection burden. Close dialogue with appropriate data holders will be a crucial aspect to successfully accomplish these goals.

In summary, these findings demonstrate the importance of including symptom propagation in models of infectious disease transmission to assist decision makers in planning infection control and mitigation strategies, where insights on epidemiological and health economic

implications of possible actions are required and where there is evidence to support the presence of symptom propagation for a given pathogen. There are still questions around whether, and to what extent, symptom propagation occurs for various pathogens and, although evidence in the literature supporting symptom propagation is accumulating, we believe it would be beneficial to reduce the uncertainty around this topic. We conclude that the consideration of symptom propagation should be commonplace in the modelling of infectious diseases and in evaluating proposed control policies from a health economics perspective. To heighten the robustness of future modelling analyses, this motivates data collection to promote the use of data-driven models and the development of analytic methods to identify the extent of symptom propagation (i.e. the value of $\alpha$) for pathogens of concern.

## Supporting information

**S1 Text. Supporting information for 'Epidemiological and health economic implications of symptom propagation in respiratory pathogens: A mathematical modelling investigation'.** This supplement consists of the following parts: (1) Methods for calculating $R_0$ and $\beta$; (2) Results with fixed $\beta$; (3) Parameterisation details of the health economic model; (4) Results for an alternative vaccine action; (5) Additional epidemiological results; (6) Sensitivity to discounting; (7) Sensitivity to intervention efficacy; (8) Additional health economic findings. (PDF)

## Acknowledgments

We thank Tom Finnie and Fergus Cumming for their helpful comments on the manuscript. We also thank Daniel Haydon, Ciaran McMonagle, Alessandro Vespignani, Jaime Earnest and Bruno Gonçalves for useful discussions of earlier versions of the ideas in this manuscript.

## Author Contributions

**Conceptualization:** Phoebe Asplin, Matt J. Keeling, Rebecca Mancy, Edward M. Hill.

**Data curation:** Phoebe Asplin.

**Formal analysis:** Phoebe Asplin.

**Investigation:** Phoebe Asplin.

**Methodology:** Phoebe Asplin, Matt J. Keeling, Rebecca Mancy, Edward M. Hill.

**Software:** Phoebe Asplin, Edward M. Hill.

**Supervision:** Matt J. Keeling, Rebecca Mancy, Edward M. Hill.

**Validation:** Phoebe Asplin.

**Visualization:** Phoebe Asplin, Matt J. Keeling, Rebecca Mancy, Edward M. Hill.

**Writing – original draft:** Phoebe Asplin, Edward M. Hill.

**Writing – review & editing:** Phoebe Asplin, Matt J. Keeling, Rebecca Mancy, Edward M. Hill.

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
