## [Decision Letter · Decision Letter 0]

27 Nov 2023

Dear Asplin,

Thank you very much for submitting your manuscript "Epidemiological and health economic implications of symptom propagation in respiratory pathogens: A mathematical modelling investigation" for consideration at PLOS Computational Biology.

As with all papers reviewed by the journal, your manuscript was reviewed by members of the editorial board and by several independent reviewers. In light of the reviews (below this email), we would like to invite the resubmission of a significantly-revised version that takes into account the reviewers' comments.

Reviewers have raised quite some important questions about how (biologically and clinically) realistic of some parameters in the model are. Authors should seriously consider these and address reviewers' concerns as provided below.

We cannot make any decision about publication until we have seen the revised manuscript and your response to the reviewers' comments. Your revised manuscript is also likely to be sent to reviewers for further evaluation.

Sincerely,

Tommy Tsan-Yuk Lam, Ph.D.

Academic Editor

PLOS Computational Biology

Thomas Leitner

Section Editor

PLOS Computational Biology

Reviewers have raised quite some important questions about how (biologically and clinically) realistic of some parameters in the model are. Authors should seriously consider these and address reviewers' concerns.

Reviewer's Responses to Questions

**Comments to the Authors:**

Reviewer #1: In this paper, the authors proposed to introduce a parameter alpha to model the symptom propagation of respiratory infections in a modified SEIR model and applied this model to three scenarios of respiratory infectious diseases, including seasonal influenza, pandemic influenza, and SARS-CoV-2. While the manuscript is well written and the methods are clearly documented, it is not clear to me how the proposed integration of the alpha parameter made a new contribution to the existing modelling framework. Please see below for my specific comments.

1. It is understandable to model the symptom propagation by a single parameter alpha ranging from 0 to 1. However, it is not clear what this alpha means clinically and how it could be applied to the design of control strategies. For example, given that there was ample data from the COVID-19 pandemic, what is the estimated alpha for Omicron BA.1 in early 2022 in the UK? If the alpha parameter were to be estimated for the Omicron outbreak, how would the control strategies be optimized?

2. Many of the results are expected. For example, on Page 6, the two beta (i.e., beta_M and beta_S) were calibrated such that the stated value of the basic reproductive number R_0 was acquired. Therefore, it is not surprising that the final cumulative infection attack rates were similar across different values of the alpha parameter. Thus, the strategies reducing symptoms or severity would be more effective with the increasing value of alpha.

3. Throughout the paper, the symptom propagation was modeled in a mechanism similar to the “all-or-nothing” mechanism in vaccines. For example, on Page 9, it was assumed that vaccinated individuals have probability eta of having mild disease and probability (1-eta) to be determined as usual severity. How would the model be modified if the symptom propagation functions in a way similar to the “leaky” mechanism in vaccines? What if there is a severity spectrum and there is not a clear cut between “mild”, “usual” and “severe”?

Reviewer #2: Asplin et al. developed a mathematical modelling framework incorporating symptom propagation and applying to a range of pathogens for investigating the epidemiological and health-economic implications of symptom propagation. The research design is appropriate and the results and methods are clearly described. I have some further suggestions that the authors could improve or discuss. Specially,

1. (Major) The authors should provide a more detailed presentation of their data source, at least, the year of data collection, the country from which the data originates, and the particular viral subtypes/clades/lineages involved. Those information can play a pivotal role in the interpretation and validity of the model's outcomes. Is there a possibility to incorporate spatiotemporal data into this study? It might provide further insights or depth to the findings.

2. (Major) Related approaches should be incorporated into this study and applied to the same dataset for the comparison.

3. Supporting Information from S12 to S21 are absent from the main text.

4. For the reproducibility, it would be helpful to have clear instructions for running the code provided in their Github repository.

5. Please indexing all the equations throughout the paper to help readers quickly locate specific equations.

Reviewer #3: The authors proposed a framework to incorporate symptom propagation in the transmission of seasonal, pandemic flu and COVID. The framework was nicely developed and presented. My main comments are about the consistency of parameters for SARS-CoV-2 variant and interpretation of the results.

Major comments:

1. Method, would the authors comment on whether the analyses should fix R0, or fix beta and let R0 to vary to draw the most meaningful conclusion?

2. There are significant differences in the epidemiological and clinical parameters between pre-omicron and omicron SARS-CoV-2 variants. Would the authors consider harmonizing all SARS-CoV-2 parameters in Table 2 and 3?

3. R0 = 3 for SARS-CoV-2 tended to be low even for pre-omicron variants

4. Severe patients are more likely to be immobile and reduce contacts with others, while very mild patients are more likely to maintain daily activities. Were the transmission parameters in Table 2 (2x and 4x transmissibility for flu and SARS-CoV-2) assumed they have same or different contact pattern?

5. Discussion, the analysis has fixed R0 and adjust beta to keep R0 constant. However, in practice the change in symptom propagation parameters may modify R0 via disease severity and corresponding contact pattern.

**Have the authors made all data and (if applicable) computational code underlying the findings in their manuscript fully available?**

Reviewer #1: Yes

Reviewer #2: Yes

Reviewer #3: Yes

PLOS authors have the option to publish the peer review history of their article (what does this mean?). If published, this will include your full peer review and any attached files.
---

## [Decision Letter · Decision Letter 1]

19 Apr 2024

Dear Asplin,

We are pleased to inform you that your manuscript 'Epidemiological and health economic implications of symptom propagation in respiratory pathogens: A mathematical modelling investigation' has been provisionally accepted for publication in PLOS Computational Biology.

Best regards,

Tommy Tsan-Yuk Lam, Ph.D.

Academic Editor

PLOS Computational Biology

Thomas Leitner

Section Editor

PLOS Computational Biology

I think that the authors have addressed all reviewers' questions.

Reviewer's Responses to Questions

**Comments to the Authors:**

Reviewer #2: The authors have adequately addressed my comments, and I have no additional suggestions.

Reviewer #3: The authors have addressed all of my comments. Thank you.

**Have the authors made all data and (if applicable) computational code underlying the findings in their manuscript fully available?**

Reviewer #2: Yes

Reviewer #3: Yes

PLOS authors have the option to publish the peer review history of their article (what does this mean?). If published, this will include your full peer review and any attached files.

Reviewer #2: No

Reviewer #3: No

---

## [Editor Report · Acceptance letter]

30 Apr 2024

PCOMPBIOL-D-23-01262R1 

Epidemiological and health economic implications of symptom propagation in respiratory pathogens: A mathematical modelling investigation

Dear Dr Asplin,

I am pleased to inform you that your manuscript has been formally accepted for publication in PLOS Computational Biology. Your manuscript is now with our production department and you will be notified of the publication date in due course.

With kind regards,

Zsofia Freund
